# Clinical Evaluation in Parkinson’s Disease: Is the Golden Standard Shiny Enough?

**DOI:** 10.3390/s23083807

**Published:** 2023-04-07

**Authors:** Foivos S. Kanellos, Konstantinos I. Tsamis, Georgios Rigas, Yannis V. Simos, Andreas P. Katsenos, Gerasimos Kartsakalis, Dimitrios I. Fotiadis, Patra Vezyraki, Dimitrios Peschos, Spyridon Konitsiotis

**Affiliations:** 1Department of Physiology, Faculty of Medicine, School of Health Sciences, University of Ioannina, 45110 Ioannina, Greece; 2PD Neurotechnology Ltd., 45500 Ioannina, Greece; 3Department of Neurology, University Hospital of Ioannina, 45110 Ioannina, Greece; 4Unit of Medical Technology and Intelligent Information Systems, University of Ioannina, 45110 Ioannina, Greece

**Keywords:** UPDRS, wearable device, motor symptoms, fluctuations, quality of life

## Abstract

Parkinson’s disease (PD) has become the second most common neurodegenerative condition following Alzheimer’s disease (AD), exhibiting high prevalence and incident rates. Current care strategies for PD patients include brief appointments, which are sparsely allocated, at outpatient clinics, where, in the best case scenario, expert neurologists evaluate disease progression using established rating scales and patient-reported questionnaires, which have interpretability issues and are subject to recall bias. In this context, artificial-intelligence-driven telehealth solutions, such as wearable devices, have the potential to improve patient care and support physicians to manage PD more effectively by monitoring patients in their familiar environment in an objective manner. In this study, we evaluate the validity of in-office clinical assessment using the MDS-UPDRS rating scale compared to home monitoring. Elaborating the results for 20 patients with Parkinson’s disease, we observed moderate to strong correlations for most symptoms (bradykinesia, rest tremor, gait impairment, and freezing of gait), as well as for fluctuating conditions (dyskinesia and OFF). In addition, we identified for the first time the existence of an index capable of remotely measuring patients’ quality of life. In summary, an in-office examination is only partially representative of most PD symptoms and cannot accurately capture daytime fluctuations and patients’ quality of life.

## 1. Introduction

Parkinson’s disease (PD) has become the second most common neurodegenerative condition following Alzheimer’s disease. Worldwide evidence highlights the rising prevalence of the disease, especially after the sixth decade [1], with an almost 10-fold increase in disease incidence from the sixth to the ninth decades of life [2]. Accounting for the predicted steep increase in PD cases until 2030 [3], the stress of upsizing on healthcare systems and the augmented burden on healthcare providers around the globe may result in system overburdens and poor patient care.

The pathogenetic cascade of PD finally results in a decrease in neurotransmitter dopamine in the caudate nucleus and putamen. This mainly involves the accumulation of Lewy bodies containing aggregates of α-synuclein with a prion-like propagation motif, leading to the loss of dopaminergic neurons in the substantia nigra pars compacta and the striatal dopaminergic denervation [4]. A clinical diagnosis is mainly based on three predominant motor symptoms as delineated by the Movement Disorder Society: bradykinesia, rigidity, and resting tremor [5]. Other motor symptoms that commonly manifest as the disease progresses are hypophonia, a decline in facial expressions, and gait impairment with freezing and falling, as well as the wearing off of responses to treatments, leading to fluctuations of the symptoms along with dyskinesia [6]. Although motor symptoms are the principal identifiers of PD, a variety of non-motor symptoms are also implicated in PD development and progression. However, these symptoms are widely under-reported by patients or are overlooked during clinical assessments.

Currently, there is no proven neuroprotective or disease-modifying therapy that can stop or delay the progress of the degeneration in PD. However, the most effective substance in terms of improving motor symptom complications with the fewest short-term adverse effects is levodopa. The systematic administration of levodopa has remained the greatest weapon in a physician’s arsenal for more than 50 years [7]. Along with other dopaminergic pharmacological targets comprising the dopaminergic therapy regime, dopaminergic medications aim to restore dopamine homeostasis at the synaptic level by transiently acting on elements implicating dopamine metabolism and neuronal excitability [8]. Over time, patients’ therapeutic window is pruned, losing their long-duration response to dopaminergic medication due to the fact of disease-related pathophysiological changes in the brain. The time without good control of symptoms (OFF) and therapy complications, such as dyskinesias, become more frequent as the disease progresses, and the effort to keep patients well controlled turns out to be really challenging [9]. Thus, questions such as “What is going to happen next?” are a common riddle that physicians are called upon to solve.

Recent care strategies include brief appointments of random frequency at outpatient clinics where, in the best case scenario, neurologists evaluate disease progression using established rating scales and/or patient-reported questionnaires, illustrating patients’ state at home. The Movement Disorder Society-Sponsored Revision of the Unified Parkinson’s Disease Rating Scale (MDS-UPDRS) is a rater-based scale globally applied in routine clinical practice and commonly accepted as a reference standard for assessing patients’ motor and non-motor symptoms and complications during a physical examination visit. Although MDS-UPDRS, as a benchmark for PD assessment, possesses comprehensive clinimetric properties, the low rater consistency and modest within-subject reliability due to the fact of psychometric issues, especially for patients’ longitudinal follow-up, create doubt in symptom evaluation [10,11]. Furthermore, the proper application and interpretation of such a scale relies on physician expertise and intuition, and these are scientific skills not widely mastered. In addition, this brief examination not only provides a snapshot of patients’ actual condition but is also subject to the Hawthorne effect, meaning that the patient performs their best during an evaluation [12].

Diaries, a useful tool for collecting self-reported feedback on the disease situation and daily activities longitudinally, are extensively applied in PD as a qualitative input to support physicians’ decision making. As a medical source of information, however, they are subject to many types of bias, creating the potential for fallacy behind physicians’ line of thinking. Mental recession is a commonly encountered situation in PD patients leading to recall bias [13]. Moreover, individuals are not duly qualified or self-aware to identify—all the more so discriminate—disease symptomatology [14]. These shortcomings in the current management of PD in combination with the relatively high number of patients (approximately 40%) that do not have access to consultation with a PD specialist or neurologist result in a higher risk of disease-related complications and mortality [15].

To address this issue, sensor-based systems have been developed over the last decade to monitor patients in their own environment and evaluate disease symptomatology [16,17,18,19]. The results of the monitoring are analyzed remotely, while a summary is provided to the treating physician and/or patient. The specialist can use this summary to assess motor and non-motor symptoms and how they are affected by the use and timing of medication. The data should be used to determine whether any changes to the treatment regimen are desirable, in consultation with the patient. These outcomes are intended to complement existing assessment methods and are not intended as substitutes. In this way, the physician has a more comprehensive picture of the patient, as well as a clear argumentation of the outcome of the treatment and the need for any modification. Of interest are the practical recommendations issued by a movement disorder specialist panel, pinpointing the value of an objective continuous monitoring strategy acting ambulatory to routine clinical practice to promote evidence-based decision making [20]. However, while many monitoring systems have been tested for their usability and accuracy in motor symptom detection [21,22,23,24,25], none of them can actually record the wide range of PD symptomatology.

Thus, in the present study we implemented a device that can accurately record almost all PD motor symptoms [26]. The question that remains is how informative the in-person clinical evaluation is compared to patients’ state at home during activities of daily living. To answer this question, we performed an observational clinical study comparing the results of the in-person clinical evaluation to the recording of symptoms in a home environment with the AI enabled wearable system PDMonitor^®^.

## 2. Materials and Methods

### 2.1. Patients

In this study, we included 20 patients who met the MDS diagnostic criteria for PD [5], ailing from a mild to moderate disease (H&Y stage from 1 to 3), aged 55–82 years, with disease duration from 1 to 22 years (Table 1). All assessments included the use of the monitoring device PDMonitor^®^, patient-reported outcome measures (MDS-UPDRS part I and II), and clinician-provided rating scales (MDS-UPDRS part III and IV) that were performed as part of routine care. A total of 21 recording periods (one patient had two recording periods) were performed using the PDMonitor^®^ system, corresponding to 81 recording days. Approval (number: 476/26-06-2022) was granted by the Ethics Committee of the University Hospital of Ioannina, and written informed consent was acquired from all patients who participated in our study to review their medical records, clinical evaluation outcomes, and diagnostic tests applied retrospectively and prospectively throughout the study’s duration. Information was made anonymous and de-identified prior to the analyses.

### 2.2. Study Process

The participants recruited in the study were consecutive patients of the outpatient clinic of neurology at the University Hospital of Ioannina, with a scheduled appointment for a routine clinical follow-up examination as part of routine clinical practice. All participants that were included in the study met the inclusion criteria (age 20–90 years and fulfilling the MDS clinical diagnostic criteria for PD) and were able to operate the device at home. An expert neurologist reviewed the patient history (medication scheme, diagnostic tests, etc.) and proceeded to the physical examination. Motor and non-motor symptoms were clinically assessed using the MDS-UPDRS part I, II, III, and IV. For patients on levodopa treatment, the examination was carried out at least one hour after the intake of the medication. Furthermore, patients were screened for possible cognitive decline using the MoCA test, and they were asked to answer a questionnaire related to their quality of life (PDQ-8). The examination visit was also video recorded to ensure data validation. A confirmatory evaluation of the patients’ condition was performed afterwards using the filmed record. Afterwards, each patient was instructed to use the PDMonitor^®^, a system designed for remote symptom monitoring to objectively evaluate a patient’s symptoms and activity in their home environment. The patients were asked to wear the device for a period of time (hereafter termed the “recording period”) determined by the physician, with 2 days as a minimum requirement of use. The physician was then able to check their condition at home by evaluating the results of a recording period using the personalized platform of the PDMonitor^®^ system (PDMonitor^®^ Physician Tool).

### 2.3. Monitoring Device

The PDMonitor^®^ system, developed by PD Neurotechnology^®^ Ltd., is a CE-marked medical device designed to be used by patients diagnosed with PD. It is intended to trace, record, process, and store a variety of symptoms frequently presented in PD. The system is composed of a base, five (5) wearable monitoring devices, and a physician web-based dashboard.

The PDMonitor^®^ consists of 5 inertial measurement unit (IMU) sensors, each of which constitutes a 9-degree measurement system (tri-axial accelerometer, gyroscope, and magnetometer). The PDMonitor^®^ symptom evaluation process transforms the raw IMU signals from all sensors into a unique set of movement features, which are in turn converted into the UPDRS, or other clinical scales’ items, estimated in 30-minute windows. This is accomplished by applying highly sophisticated algorithms that first identify different areas of activity and then detect, distinguish, and quantify PD symptoms and conditions, mainly focusing on dyskinesia, bradykinesia, gait, tremor, posture detection, FoG, ON/OFF fluctuations, and activity. The PDMonitor^®^ provides an estimation in the UPDRS for the majority of the symptoms, except leg tremor, FoG, and postural instability. So, machine learning is used for the detection of activity and presence of symptoms, while for the estimation of symptom severity more deterministic methods such as linear regression models are employed. The main outcomes of the PDMonitor^®^ are presented in the table below (Table 2), and the methodology and accuracy of the system have been extensively described in a recent paper [26].

Moreover, the PDMonitor^®^ classifies as activity any motion other than tremor and dyskinesia. The activity can be further subclassified on indicators expressing the rate of time a patient is in a state of general immobility or general activity and whether the patient is in a sitting or lying position. The PDMonitor^®^ dUPDRS part III, in fact, resembles the UPDRS total part III score, digitalized, in the context of cumulating the score of the individual symptoms. We used the dUPDRS (digital-UPDRS) definition conventionally to distinguish the PDMonitor^®^ measurement from that of the actual scale used as a benchmark.

The patients were instructed to wear the monitoring devices at specific positions of the body (2× shanks, 2× wrists, and 1× waist) during the awake hours of a day. They were instructed to remove them only when they took a shower or in the case of intense activity, e.g., playing sports. The patients’ motor symptoms and therapy complications were then graphically presented to healthcare professionals in an easily interpretable way via the web dashboard. The physicians could open the report at any time and scrutinize the course of the disease. An example of a PDMonitor^®^ report derived from a study case is presented below (Figure 1).

By doing so, this integrated ecosystem assists physicians in deciding on a therapeutic plan and in being aware of a patient’s prejudicial habits. Furthermore, this easy-to-use system [29] has already demonstrated considerable accuracy in the assessment of dyskinesia, tremor, and freezing of gait [24,30].

### 2.4. Data Analysis Methods

Statistical analysis was conducted using Pearson’s correlation test to examine the association between the PDMonitor^®^ data and the clinical scale scores. The Bland–Altman plot method was used to evaluate the agreement of the two assessment methods on selected symptoms or disease state. The symptom values with a correlation coefficient value of less than 0.6 were not tested for agreement, as the correlation was considered low in terms of clinical significance. All data are displayed as the means (±SD) or frequencies (%), and the significance level was set at *p* < 0.05 (two-tailed test). We used commonly accepted coefficient (r) cutoff points to measure the strength of the relationship between the two measurement methods (PDMonitor^®^ and physical examination). When the coefficient magnitude is between 0 and 0.10, the correlation is considered negligible; likewise, when the calculated coefficient (with a positive or negative sign) is between 0.10 and 0.39, 0.40 and 0.69, 0.70 and 0.89, and 0.90 and 1.00, then the relationship is considered “weak”, “moderate”, “strong”, and “very strong”, respectively [31].

## 3. Results

The results describing the degree of agreement between the physical examination and remote monitoring measures are analyzed in detail throughout the following subsections. The correlations are presented in correlation plots, and for the symptoms with relevant concordance, the agreement of the methods is presented using Bland–Altman plots.

### 3.1. Symptom Evaluation

#### 3.1.1. Bradykinesia

In this section, we examine the correlation of the PDMonitor^®^ with the MDS-UPDRS score for bradykinesia. The MDS-UPDRS bradykinesia scores used for this analysis included data from finger tapping (3.4), hand movement (3.5), and pronation/supination (3.6), which were compared separately and in total with the PDMonitor^®^ bradykinesia scores (hereafter PDM-BRAD when referring to the aggregated score for both hands). The analysis includes data from patients who did not present severe dyskinesia during the visit to such an extent that it would affect the assessment of symptoms (as imposed by the instructions of the MDS-UPDRS). This resulted in data from two patients being exempted. A moderate correlation was observed between the at-home bradykinesia assessment and the in-person clinical examination, as summarized in Table 3.

Particularly, the correlation coefficients (r) for finger taps, hand movements, and pronation supination ranged from 0.51 to 0.63 (*p* < 0.001). Furthermore, after analyzing the correlation between the PDMonitor^®^ and the aggregated sum for both hands, we found a Pearson coefficient of r = 0.62 (*p* < 0.001), with good agreement expressed by a bias of 0.20 (Figure 2).

Comparing the PDMonitor^®^ data on bradykinesia across the different disease stages, we found that the moderate-to-late stage group had significantly (*p* < 0.05) higher bradykinesia scores than the early-stage group (Figure 3), indicating the consistency in recording bradykinesia both at the visit and during home monitoring at different stages of the disease.

#### 3.1.2. Rigidity

Rigidity is a symptom that cannot be evaluated directly by wearable systems. Thus, we compared the MDS-UPDRS rigidity score with the PDMonitor^®^ values for bradykinesia, tremor, gait, and OFF. Strong and moderate correlations were found (Table 4) between all these PDMonitor^®^ values and the rigidity of the physical examination. The Pearson coefficients of 0.54, 0.74, 0.51, and 0.61 signify the correlation among the total average score of all five body parts, as evaluated by the clinical scale item 3.3 (rigidity) and the PDMonitor^®^ average values on bradykinesia, tremor, gait, and OFF, respectively (Figure 4).

#### 3.1.3. Tremor

To evaluate any possible concordance between the clinical scale and the PDMonitor^®^ results for the presence of tremor, we compared the value from MDS UPDRS item 3.17 (except jaw metric) for the upper and lower extremities with the PDMonitor^®^ values for the corresponding body parts. Two evaluations were excluded from the analysis of the severity of the resting tremor, as severe dyskinesias affected the assessment of tremor during the clinical evaluation. Thereafter, we attempted to correlate both measurement methods to detect resting tremor constancy (item 3.18) for the upper and lower extremities. The results are also presented as the aggregated sum of both the upper and lower extremities. When analyzing for upper extremities by using both the individual values of the MDS-UPDRS item 3.17 and the 3.17 aggregated value, we found strong correlation with PDMonitor^®^ values for resting tremor. However, we found weak to medium correlations with Pearson coefficients (r) of 0.23 and 0.58 for the right and left leg, respectively. The results are summarized in Table 5. When reaching for the agreement on detecting resting tremor of the arms (aggregated values), we found that the PDMonitor^®^ values were normally distributed between the limits of agreement, as presented in the Bland–Altman plot (Figure 5d). The average bias found indicates that the PDMonitor^®^ measures 0.37 units less than MDS-UPDRS.

We then conducted an analysis on the correlation of the resting tremor constancy. The correlation was based on the score of item 3.18 of the MDS-UPDRS and the relevant PDMonitor^®^ value. To make the comparison, it was necessary to adjust the result of the PDMonitor^®^ tremor value recording (rate of resting tremor during the day) to a numerical integer rating, such as those of the MDS-UPDRS (0–4). The strong correlation found (r = 0.76, *p* < 0.001) demonstrates that in addition to assessing the severity of resting tremor, tremor constancy, as assessed during the visit, is a representative indicator of what is occurring at home. The correlation for the tremor constancy of the lower limbs was not performed. Apart from the small number of patients who had tremor in the lower limbs (based on clinical examination), those patients also had coexisting tremor in the hands of equal or greater severity. Therefore, the perception of tremor duration was determined and evaluated mainly by the occurrence of tremor in the hands.

#### 3.1.4. Gait and Balance

A moderate correlation was found on gait evaluation (r = 0.49, *p* < 0.05) by comparing item 3.10 of the MDS-UPDRS during the in-office clinical evaluation with the composite PDMonitor^®^ measure of gait in the home environment, an informative index signifying the overall impairment in gait biomechanics. The correlation between the PDMonitor^®^ gait and patient-rated gait disturbance described by UPDRS item 2.12 was found to be of moderate grade (r = 0.63, *p* < 0.01) (Figure 6b).

Similarly, a moderate correlation was identified on detecting events of posture instability (r = 0.46, *p* < 0.05) by comparing the presence of instability, as evaluated by performing the “pull test” (item 3.12, score ≠ 0), with the PDMonitor^®^ value, indicating at least one probable instability event during the recording period. Despite the moderate correlation in the detection of instability events and gait impairment, a very strong correlation was found (r = 1) on freezing of gait event detection by comparing the presence of FoG, as evaluated from the in-person examination (item 3.11, score ≠ 0), with the PDMonitor^®^ value, indicating at least one probable FoG event during the recording period.

Furthermore, the correlation between the two measurement methods on the acuteness of the gait impairment and FoG was also investigated. A strong correlation was identified between the PDMonitor^®^ FoG severity arbitrary score and the clinical scale quantitative ratings from both physician (r = 0.78, *p* < 0.001) and patient (r = 0.74, *p* < 0.001) on FoG (items 3.11 and 2.13), as presented in Figure 7.

Hence, a strong to very strong correlation was identified on detecting symptom harshness and FoG events, respectively.

#### 3.1.5. Total Score of Motor Examination

A moderate correlation was detected for the dUPDRS part III total score, as calculated by the PDMonitor^®^ (mean score), and during the in-person examination (r = 0.48, *p* < 0.05). To calculate the correlation in patients who were OFF during the clinical examination, the maximum PDMonitor^®^ value was used rather than the mean (Figure 8).

### 3.2. Motor Complications

#### 3.2.1. Dyskinesias

Strong correlation (r = 0.75, *p* < 0.001) is found in the percent of time patients are experiencing dyskinesia during the awake hours of the day between in-person examination and PDMonitor^®^ recordings at home, with mean difference detected of only 1% with clinical scale lagging behind PDMonitor^®^ (Figure 9). Accordingly for the detection of dyskinesia we compare the percent of time as calculated in item 4.1 with the corresponding aggregated measure of PDMonitor^®^.

#### 3.2.2. OFF Time and Distribution of OFF Severity across Different Stages

A moderate correlation was found on the detection of OFF time (r = 0.65, *p* < 0.01) between the two measurement strategies (Figure 10a). A strong correlation was identified (r = 0.77, *p* < 0.001) for the functional impact that the OFF condition had on the patient (Figure 10b). For the detection of the OFF time and OFF severity, we compared the percent of time and the functional impact of OFF as calculated in items 4.3 and 4.4 of the MDS-UPDRS part IV with the corresponding measures of the PDMonitor^®^.

As shown in Figure 1, the PDMonitor^®^ can capture the severity of a symptom or condition over time. Therefore, we also evaluated patients’ OFF severity in different disease stages. In Figure 10c, it is shown that the severity of the OFF time increases with the disease stage.

### 3.3. Disease Implication on Patients’ Quality of Life

Two types of questionnaires were used in this study, which largely reflected the impact of the disease on patients’ daily activities and general quality of life. These included part II of the MDS-UPDRS and the disease-specific PDQ-8. These two instruments showed a strong correlation (r = 0.71, *p* = 0.001).

The percent of time a patient was OFF, as measured by the PDMonitor^®^, appeared to have a moderate correlation with the total score of part II (r = 0.49, *p* < 0.05). Moreover, the same PDMonitor^®^ index strongly correlated with the PDQ8 total score (r = 0.73, *p* < 0.001). Of the eight domains, the one with the greatest correlation with the PDMonitor^®^ OFF time was the mobility domain (r = 0.68, *p* < 0.01), followed by concentration issues (r = 0.60, *p* < 0.01). The results are summarized in Figure 11.

In addition to the ability to detect PD-related motor symptoms, the PDMonitor^®^ determines activity levels. The activity indicators express the rate of time a patient is in a state of general immobility or general activity and whether the patient is in a sitting or lying position. Patients in the early stages of the disease showed higher activity than those in more advanced stages (with an average percent of being active of 32% vs. 39% for the H&Y stages > 2 and ≤2, respectively), indicating a better quality of life for the former.

## 4. Discussion

The routine clinical practice currently followed in Greece is streamlined with most countries worldwide, comprised of infrequent, brief appointments at outpatient clinics. Oftentimes, non-PD expert physicians try to apprehend motor symptoms along with complications and intuitively track changes outside of the clinic with some ecological validity. However, the tools used in clinical practice to monitor temporal disease patterns in symptom fluctuations have various limitations. Symptom diaries, validated clinical scales, such as the MDS-UPDRS, and patient-reported questionnaires, such as QoL questionnaires, are often retrospective, being subject to recall bias and lack objectivity, or suffer from inter- and intra-rater variability [10,13]. Symptoms are very often overlooked by raters or under-reported by patients. Therefore, objective measurements providing an accurate evaluation of symptoms can pave the way for evidence-based clinical decision making [32].

Up-to-date AI-driven sensor-based technologies still lack concrete evidence of their clinical benefit. However, favorable for their adoption into routine practice are the results of the diagnostic accuracy for most of the devices on the market [21,22,23,25]. To the best of our knowledge, only a few studies using remote monitoring devices reported association outcomes, correlating home monitoring with visit assessment using expert-response clinical scales. Bradykinesia, dyskinesia, and tremor scores in a home environment were generally moderately correlated with MDS-UPDRS scores for in-office clinical evaluations [33,34,35].

In the present study, we aimed to evaluate how comprehensive the information provided by routine visits to outpatient clinics for motor symptoms is in relation to the actual situation of patients at home, as reflected by a wearable remote monitoring medical device. The comparative analysis of the results focused on two axes. The first investigated the correlation of the motor symptoms detected in the clinical evaluation with the MDS-UPDRS to the objective measurements of the PDMonitor^®^ system. In the second, correlations between the system’s recordings and the patients’ quality of life were explored.

All motor symptoms of the patients with PD participating in the study were evaluated by means of in-person clinical examination with the MDS-UPDRS, as well as in-house monitoring with the PDMonitor^®^. Comparing the results, we appreciate that there were various degrees of agreement between the two methods for all symptoms, ranging from moderate to strong correlations (Table 6). This shows that the clinical examination of the patients with PD, even when performed with validated scales, can only partially reveal the situation of the patients in their home environment during their daily activities.

For bradykinesia, rigidity, gait impairment, and instability, there was a moderate correlation between the clinical evaluation and the recordings in the home environment. The differences observed could possibly be attributed to the effort that patients usually exert during their visit at the physician’s office. These specific symptoms can be partly masked during the patient’s “best performance” in front of the treating physician, while they are quite differently expressed during “usual performance” in activities of daily living. On the other hand, for freezing of gait, tremor, and dyskinesia, a high correlation was revealed, except rest tremor in the lower limbs. The latter can be explained by the fact that the symptom does not recur as often in the lower as in the upper extremities. Not all patients with tremor in the lower limbs experienced the symptom and, in particular, the limited duration of the in-person evaluation reduced the likelihood of detecting the symptom; however, the information was provided to the physician by the home recordings. These results should not come as a surprise, since even the best performance cannot overcome the sudden inability to step forward or involuntary movements.

Intriguingly, although the device has not yet demonstrated any surrogate marker of rigidity, a moderate correlation was observed for bradykinesia, gait, and %OFF as measured by the device. In addition, the PDM-TREMOR was strongly associated with the average rigidity score. This does not necessarily imply the establishment of a composite measure of rigidity, as this cardinal symptom occurs and aggravates at a similar rate to bradykinesia and gait and balance disturbances but not at the same rate as tremor [36]. Nevertheless, the strong correlation with the PDMonitor^®^ tremor offers a chance to study this association in the future, especially considering that the majority of patients participating in the study were tremor-dominant (14/20).

Thus, the strong correlation that was observed in the identification of freezing of gait, resting tremor, and dyskinesia means that when these symptoms are detected in the clinical evaluation, they are also present in the home environment, as expected. On the other hand, for gait impairment, instability, and bradykinesia, the moderate correlation between the results of the clinical examination and the home recordings is indicative of the symptoms’ wide fluctuations throughout each day, leading to the overestimation or underestimation of a patient’s overall situation by the clinical assessment. It is interesting, however, that a symptom that cannot be captured by inertial sensors (i.e., rigidity) is also moderately correlated to the bradykinesia of the home recordings. This is another finding highlighting the value of monitoring devices for diverse motor symptom detection.

Furthermore, the clinical assessment and the results of the home recordings appear to have a moderate correlation in identifying the percentage of time in which the patients were OFF. This finding, although expected, adds great value to the home monitoring, as it is the only reliable data that the treating physician can use to adjust the treatment more effectively. By exploiting not only the temporal spread of the OFF intervals within the day but also the severity of the OFF as provided by PDMonitor^®^, physicians can accurately capture the actual state of a patient throughout a day. In this way, the physician gains a comprehensive picture of a symptom’s daily fluctuations, pinpointing the value of continuous monitoring in the management of patients with Parkinson’s disease. A summary of the correlation analysis between the PDMonitor^®^ and the MDS-UPDRS scores is presented in Table 6.

The question that arises from all of the above findings is whether there is any measure that mirrors the actual state of the patient at home and, furthermore, their quality of life. Part II of the MDS-UPDRS shows the impact of the disease on the patient’s daily activities, and it is considered a reliable measure of the patient’s quality of life. Similarly, the length of time the patient is in the OFF state is an indirect indicator of quality of life, as in this state, the patient’s overall activity and mobility are limited. The remarkable correlation of the percentage of time patients are in the OFF state, as measured by the PDMonitor^®^, with the PDQ8 scale reflects the value that artificial-intelligence-enabled recording systems have in the overall evaluation of a patient’s actual state at home. This value is further highlighted by the strong correlation between MDS-UPDRS part II and PDQ8, a result in line with previous studies [37], showing that even a detailed history taken from a patient oftentimes does not reveal their quality of life in the way that the monitoring of symptoms in their home environment can.

As of now, few studies have been undertaken comparing similar devices worn in an uncontrolled environment with accredited scales and questionnaires, e.g., UPDRS and patient diary [33,34,35,38]. The correlation between the dyskinesia score derived from a wrist-worn monitoring device (Parkinson’s Kinetrigraph^TM^) and the UPDRS part IV dyskinesias was proven to be low (r = 0.38, *p* = 0.09), while no significant correlation was found for the device bradykinesia score with the UPDRS part IV fluctuations (r = 0.25, *p* = 0.28). However, a moderate association was found between patients’ perception of disability due to the presence of motor fluctuations and the device’s fluctuation score (r = 0.52, *p* = 0.018) [35]. Recently, L. Chen et al. [34] demonstrated a moderate association between the device’s bradykinesia score and the UPDRS items indexing arms bradykinesia (items 23–25), with the coefficients (r) ranging from 0.456 (for finger taps) to 0.557 (for hand movement). Moreover, the device metrics were moderately correlated to the average score of all bradykinesia items (0.588). The percent of time with tremor, as measured by the device, showed a slender correlation with patient-reported tremor (0.269) and a moderate correlation with the expert-rated items of UPDRS part III (0.434). Of interest, the rigidity score was moderately correlated to the bradykinesia score (0.479). Notwithstanding the moderate agreement of the above measurement strategies, no correlation was present between dyskinesia, as measured by the device, and the OFF time, as reported by patients.

Furthermore, a moderate concordance was found between a waist-worn device (STAT-ON^TM^) and patient diary (Hauser diary) on the percentage of daily time in the OFF state (r = 0.57), for the percentage of daily time in the ON state (r = 0.48), and for daily time with dyskinesias (r = 0.65) [33]. The gait and balance characteristics measured using another wearable device (OPAL), composed of three wearable sensors worn at the lumbar and on both lower extremities, demonstrated a moderate association with total UPDRS part III, postural instability and gait difficulty subscores of UPDRS part III, and rigidity subscore with coefficients of 0.48, 0.61, and 0.49, respectively [38]. This benchmarking is not, of course, a compendium, as the context in which each technology is used, the methods of comparison, and the scales employed vary from one study to another. However, each study shares the same goal, namely, to demonstrate the clinical validity and degree of association between the remote objective monitoring of a patient’s symptoms in a free-living setting and the golden standard of in-person clinical examination using validated clinical scales.

The present study confirms the results of previous studies regarding the degree of correlation in identifying the main symptoms of the disease (rigidity and bradykinesia) and disease complications (dyskinesia and ON/OFF fluctuations) with improved rates, however. Notably, for resting tremor we found a strong correlation both in terms of severity and constancy, not reported in other studies to such a high degree. Moreover, the moderate and high correlation in the detection of OFF and dyskinesia, respectively, are variables of particular clinical relevance in terms of their impact on medication adjustment, underlining at the same time the reliability of clinical assessment and the added value that remote monitoring can have in this context. Interestingly, a strong correlation was found between the OFF rate detected by the device and the PDQ8 on the patients’ quality of life. The existence of an index capable of remotely measuring patients’ quality of life based on the objective recording of their motor status and symptoms is an innovation that may contribute to a more effective management of the disease.

## 5. Conclusions

Wearable systems are increasingly being implemented in everyday clinical practice for PD, supporting medical decisions alongside in-person clinical examination. Most studies have pointed to the use of such technologies for diagnostic confirmation and to monitor PD progression from its initial stages [39]. Our study’s results demonstrate the value of using sensor-based systems such as PDMonitor^®^ for the better understanding of PD symptoms and their variations during the day. This is a situation that will eventually put patients and healthcare experts as beneficiaries by promoting patients’ health literacy and clinicians’ quality of care in order to pursue optimal personalized treatments. Overall, it could be argued that the examination at the doctor’s office is still partly representative for most PD symptoms and cannot accurately capture the fluctuations during the day and the patient’s quality of life. Of accruing interest would be the monitoring of patients over longer periods to control the intermediate impact of monitoring applied individually in the context of routine patient treatment, particularly in groups of patients with motor fluctuations, patients who do not respond to treatment as expected, levodopa-naive patients, and patients referred for advanced therapies.

## Figures and Tables

**Figure 1 sensors-23-03807-f001:**
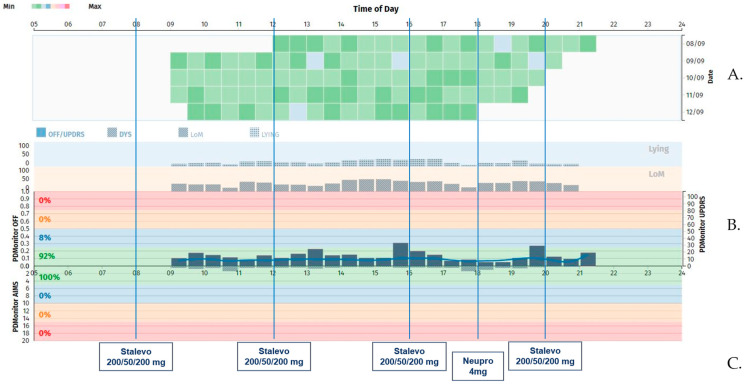
PDMonitor^®^ OFF/dyskinesia chart for a well-controlled patient with slight fluctuations: (**A**) severity of a symptom for a 30 min interval using a heat map displaying the symptom severity; (**B**) chart with the average symptom intensity for any time of day (bars above the x-axis represent OFF while below the presence of dyskinesia); (**C**) the medication the patient receives.

**Figure 2 sensors-23-03807-f002:**
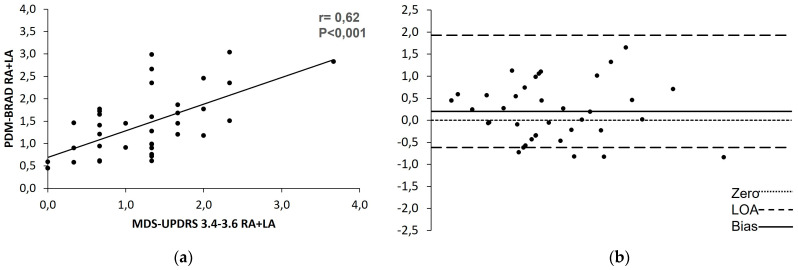
The correlation on bradykinesia between the PDMonitor^®^ and the aggregated clinical scale scores using (**a**) Pearson’s correlation and (**b**) the Bland–Altman test. LoA, level of agreement; d-line, mean difference; PDM-BRAD, aggregated bradykinesia recorded by the PDMonitor^®^ for both arms.

**Figure 3 sensors-23-03807-f003:**
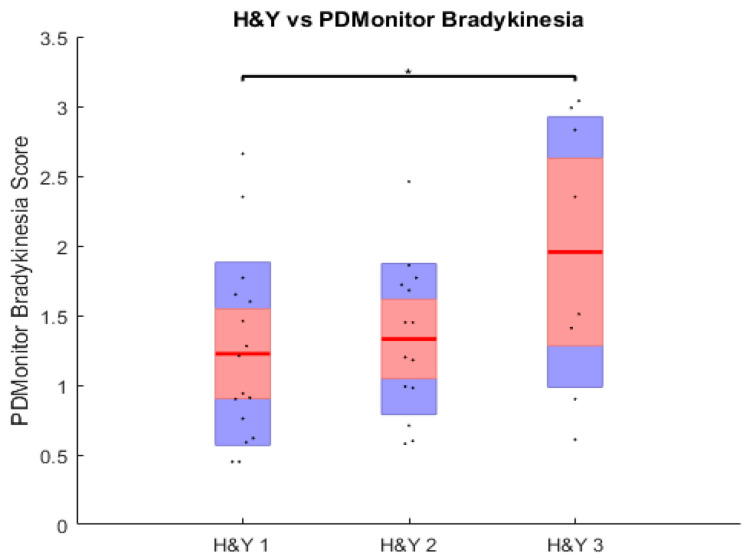
PDMonitor^®^ bradykinesia measures at different disease stages (H&Y 1–3).

**Figure 4 sensors-23-03807-f004:**
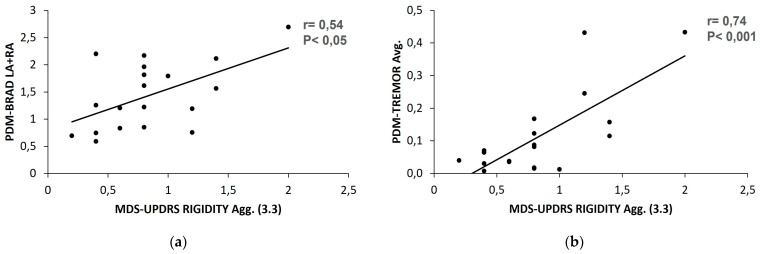
The correlation between the PDMonitor^®^ average values and MDS-UPDRS item 3.3 average score on (**a**) bradykinesia; (**b**) tremor; (**c**) gait; (**d**) %OFF.

**Figure 5 sensors-23-03807-f005:**
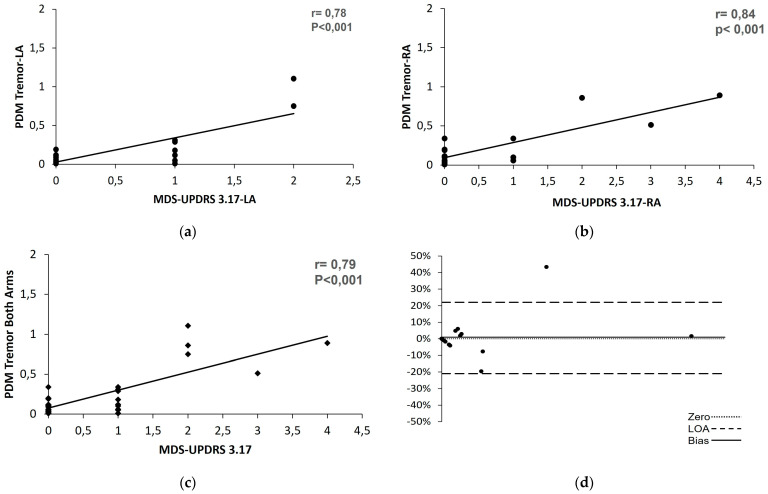
The correlation and agreement of the PDMonitor^®^ and MDS-UPDRS on resting tremor of the (**a**) left arm, (**b**) right arm, (**c**) aggregated values of both arms, and (**d**) Bland–Altman plot of the aggregated values.

**Figure 6 sensors-23-03807-f006:**
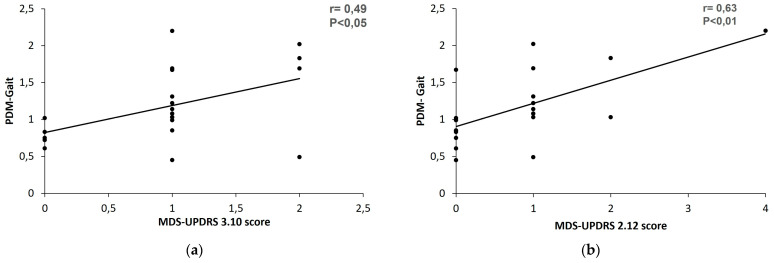
Correlation on gait between the PDMonitor^®^ score and the (**a**) MDS-UPDRS scores of item 3.10 (rater scored) and the (**b**) MDS-UPDRS scores of item 2.12 (patient scored) using Pearson’s correlation test.

**Figure 7 sensors-23-03807-f007:**
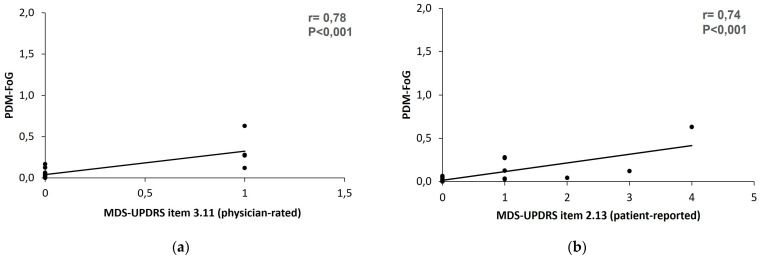
The correlation between the PDMonitor^®^ and MDS-UPDRS on the FoG (**a**) device score and physician-rated item 3.11 and (**b**) device score and patient-reported item 2.13.

**Figure 8 sensors-23-03807-f008:**
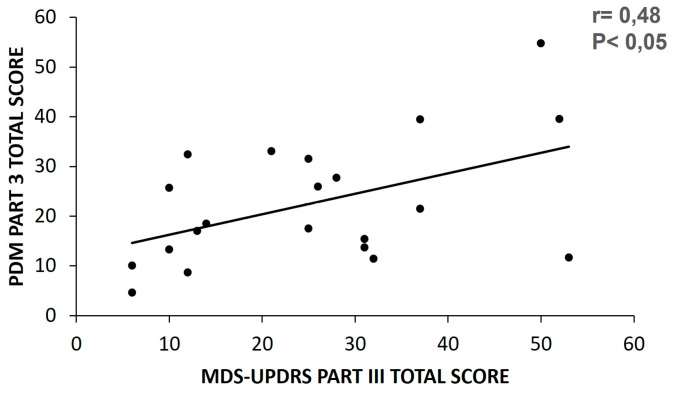
Correlation plot of the total part III score between the clinical scale and the PDMonitor^®^ (PDM dUPDRS part III).

**Figure 9 sensors-23-03807-f009:**
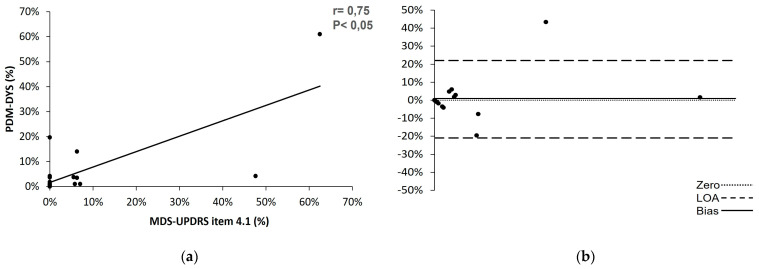
The correlation between the PDMonitor^®^ and the MDS-UPDRS for dyskinesia using a (**a**) Pearson correlation plot and (**b**) Bland–Altman test. DYS, percent time on dyskinesia as recorded by the PDMonitor^®^

**Figure 10 sensors-23-03807-f010:**
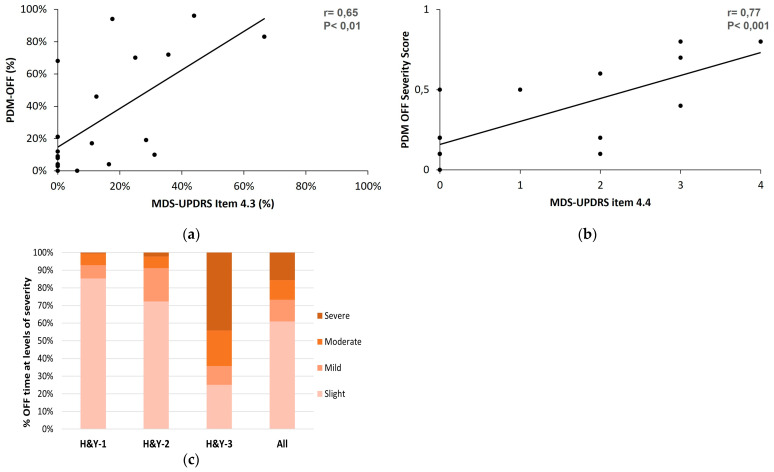
The correlation plots between the PDMonitor^®^ and the MDS-UPDRS using the Pearson’s correlation of (**a**) PDMonitor^®^ percent of time OFF and MDS-UPDRS item 4.3; (**b**) PDMonitor^®^ OFF severity score and MDS-UPDRS item 4.4 score; (**c**) the %OFF time at severity levels.

**Figure 11 sensors-23-03807-f011:**
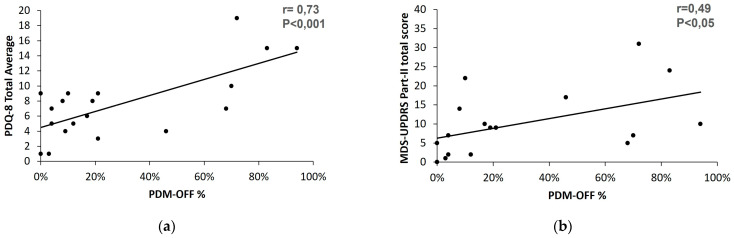
Correlation plots of the PDMonitor^®^ percent time OFF with the (**a**) total average score of the PDQ-8 questionnaire and (**b**) part II total score.

**Table 1 sensors-23-03807-t001:** The clinical- and PDMonitor^®^ -related characteristics of the participants.

Domains	Data **	Values *
Demographics	Patients (M/F)	20 (9/11)
	Age (yrs)	68.8 ± 7.7
	Disease duration (yrs)	6.1 ± 5.7
	Years with levodopa	4.0 ± 3.9
	H&Y stage	1–3
	MoCA	22.4 ± 3.8
MDS-UPDRS	Part I	11.6 ± 6.1
	Part II (M-EDL)	10.2 ± 8.5
	Part III	25.3 ± 14.7
	Part IV	3.6 ± 4.6
	Bradykinesia (item 3.4–3.6)	1.22 ± 0.70
	Resting tremor (item 3.17)	0.27 ± 0.38
	Gait (item 3.10)	0.95 ± 0.67
	Rigidity (item 3.3)	0.84 ± 0.45
PDMonitor^®^	BRAD	1.42 ± 0.73
	TREMOR	0.11 ± 0.13
	GAIT	1.17 ± 0.50
	%OFF	31%
	%DYS	6%
	dUPDRS part III	22.5 ± 12.6

* All data are displayed as the means ± SD or frequencies (%). ** M/F, male/female; H&Y, Hoehn–Yahr stage; MoCA, Montreal Cognitive Assessment; UPDRS, Unified Parkinson’s Disease Rating Scale; UPDRS I, part I of the UPDRS; UPDRS II, part II of the UPDRS; UPDRS III, part III of the UPDRS; resting tremor, total average from all body parts except jaw; PDM-BRAD, PDMonitor^®^ arms bradykinesia aggregated score; PDM-TREMOR, PDMonitor^®^ tremor aggregated score from all body parts; PDM-GAIT, PDMonitor^®^ gait score; DYS, average PDMonitor^®^ percent time dyskinesia; OFF, average PDMonitor^®^ percent time OFF; M-EDL, motor experiences of daily living; dUPDRS part III, average PDMonitor^®^ score of all motor symptoms except dyskinesia.

**Table 2 sensors-23-03807-t002:** PDMonitor^®^ outcomes and the respected measurement scales correlated and methods followed.

PDMonitor^®^ Outcome	Measurement Scale/Method
Bradykinesia (right and left arm)	Average of UPDRS items 23–25
Wrist tremor (right and left)	Wrist tremor amplitude estimation using a fuzzy linear function to correlate to the score of UPDRS item 20 (arms)
Leg tremor (right and left)	Activity detection method, where the activity classified as tremor is correlated to the UPDRS item 20 (legs) score
Gait impairment	The estimation is based on gait analysis, where the range of motion is calculated and translated into the UPDRS item 29 score
Freezing of gait	Evaluation of FoG events by applying the freezing index introduced by Moore et al. during pausing phases [27]
Postural instability	Device estimate of the swing time variability of the lower extremities. The possibility of an instability event occurring is presented on a scale ranging from 0 to 1
Time spent with dyskinesia (%)	The percent of time with dyskinesia over a threshold determined by expert annotation based on the AIMS scale
Time spent in OFF state (%)	OFF time estimation is based on the relief method which, after combining each symptom and measure individually, estimates the importance of each symptom [28]
PDM-dUPDRS part III	PDM estimation of the part III score based on a regression model of the individual symptoms of bradykinesia, tremor, gait, FoG, and instability converted to a UPDRS sum score

**Table 3 sensors-23-03807-t003:** The correlation of the MDS-UPDRS scores on the bradykinesia items to the PDM-BRAD score using Pearson’s coefficient.

MDS-UPDRS Item vs. PDM-BRAD	Coefficient (r)	*p*-Value
Aggregated bradykinesia (3.4–3.6)	0.62	<0.001
Finger taps (3.4)	0.63	<0.001
Hand movement (3.5)	0.52	<0.001
Pronation supination (3.6)	0.51	<0.001

**Table 4 sensors-23-03807-t004:** The correlation of the MDS-UPDRS rigidity score (total average) to the PDMonitor^®^ average values on bradykinesia, tremor, gait, and %OFF using Pearson’s coefficient.

MDS-UPDRS Item 3.3 Avg. vs. PDM	Coefficient (r)	*p*-Value
PDMonitor^®^ BRAD	0.54	<0.05
PDMonitor^®^ TREMOR	0.74	<0.001
PDMonitor^®^ Gait	0.51	<0.05
PDMonitor^®^ %OFF	0.61	<0.01

**Table 5 sensors-23-03807-t005:** The correlation of the MDS-UPDRS scores on resting tremor severity and constancy to the PDMonitor^®^ tremor using Pearson’s correlation.

MDS-UPDRS Item vs. PDMonitor^®^ Tremor	Body Part	Coefficient (r)	*p*-Value
Resting tremor severity (3.17)	Both arms	0.79	<0.001
	Left arm	0.78	<0.001
	Right arm	0.84	<0.001
	Both legs	0.33	<0.05
	Left leg	0.58	<0.01
	Right leg	0.23	>0.05
Resting tremor constancy (3.18)	Both arms	0.76	<0.001
	Left arm	0.65	<0.001
	Right arm	0.87	<0.001

Both arms and both legs, as well as the sum of the values for both extremities (upper and lower).

**Table 6 sensors-23-03807-t006:** Summary of the correlation of the MDS-UPDRS items to the PDMonitor^®^ metrics.

MDS-UPDRS Item	vs. PDMonitor^®^	Coefficient (r)	*p*-Value
Bradykinesia (3.4–3.6)	vs. PDM-BRAD	0.62	<0.001
Rigidity (3.3)	vs. PDM-BRAD	0.54	<0.05
	vs. PDM TREMOR	0.74	<0.001
	vs. PDM Gait	0.51	<0.05
	vs. PDM OFF%	0.61	<0.01
Arms resting tremor (3.17)	vs. PDM arms tremor	0.79	<0.001
Arms resting tremor constancy (3.18)	vs. PDM tremor constancy	0.76	<0.001
Gait (3.10)	vs. PDM gait	0.49	<0.05
Gait—patient rated (2.12)	vs. PDM gait	0.63	<0.01
FoG presence (3.11)	vs. PDM FoG event	1	-
FoG severity (3.11)	vs. PDM FoG severity	0.78	<0.001
FoG—patient rated (2.13)	vs. PDM FoG severity	0.74	<0.001
Instability presence (3.12)	vs. PDM instability event	0.46	<0.05
Dyskinesia (4.1)	vs. PDM dyskinesia%	0.75	<0.001
%OFF (4.3)	vs. PDM OFF%	0.65	<0.01
Functional impact of OFF (4.4)	vs. PDM OFF severity	0.77	<0.001
Total part III	vs. PDM dUPDRS part III	0.48	<0.05
Total part II	vs. PDM OFF%	0.49	<0.05

## Data Availability

The raw data concerning questionnaires and scales supporting the conclusions of this article can be made available by the authors upon reasonable request.

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
