# Peer review of "Clinical Evaluation in Parkinson’s Disease: Is the Golden Standard Shiny Enough?"

_sensors, 2023, doi:10.3390/s23083807_

Round 1

Reviewer 1 Report (Previous Reviewer 1)

I can change my review to minor revision with following comment:

Fonts in figures are too small. Please increase them for better readability.

Author Response

Thank you for the comments.

All fonts in figures were increased in the new version of the article to a more readable size.

Reviewer 2 Report (Previous Reviewer 3)

Thank you to the author team for providing responses to each of the comments and making subsequent changes. I feel the manuscript reads well now with the changes. In particular the additions to the methods section have been really vital. I would be happy to recommend this useful piece of work after a few minor revisions and would very interested to use use PD monitor in my own clinical practice one day.

1) Recommend the acronym IMU is expanded. The acroynm MD (line 166) is not clear to me and should also be expanded (is it movement devices?)

2) Thank you for expanding on the methods which has helped considerably. Table 2 is useful for understanding the individual outputs, but I am still not certain what dimension these take. Is the a value from 0-4 on a similar scale clinical UPDRS score? I think this is what is being suggested based on the line "provides estimation in UPDRS scales" but it is vital to make sure this is really clear for the reader.

3) In table 2 the final PDmonitor outcome is  "UPDRS part III". It is stated that this is calculated based on a regression model of individual symptoms, however it would be important to know exactly which components this uses -is this all components including off time, off severity, or just certain ones? I think it is also potentially a bit misleading to call this "UPDRS part III" in the table because it is an estimation based on an aggregate accelerometry score, not the aggregate of the validated clinician determined score and should be designated as something potentially similar but not the same- eg. "eUPDRS score" (e for estimated/electronic). This is also referred to variably in the results section including "PDM Part 3 total score" (figure 8) or just "part III total score" (line 332). Care should be taken here to use the term consistently. Finally, UPDRS is copyrighted by the Movement Disorder Society which could cause problems if used without permission?

3) On table 2 it is really quite hard to work out which scale goes with each outcome. Suggest a gap between different measurement factors for each PDmonitor outcome category 

4) There is a reference software error line 279

Author Response

Thank you very much for the comments. We have made the following changes in the new version of the manuscript, according to these comments.

1) The IMU acronym was expanded, and MD acronym was changed to ‘’sensors’’.

2) A clarification on the method of calculating the scale has been added to the corresponding fields of the methods section. Briefly, to calculate the overall UPDRS Part III score (renamed PDM dUPDRS part III), the sum of all symptoms calculated by the system is used, which is then converted to the UPDRS scale score. Additional adaptations of the definition (dUPDRS) were performed in section 3.1.5.

3) The “PDMonitor estimated UPDRS part III” score has been changed to “PDM dUPDRS part III” (digital UPDRS) throughout the text and the figures.

4) Table 2 was updated according to the comments.

5) The error has been corrected.

This manuscript is a resubmission of an earlier submission. The following is a list of the peer review reports and author responses from that submission.

Round 1

Reviewer 1 Report

Summary: The paper presents an evaluation of a wearable device and its usefulness in obtaining PD evaluations in a out of clinic setting. Experiments with 20 patients show that measurements from the wearable device have high correlation with clinical results.

Comments:

- Evaluations with 20 patients is promising and shows the relevance of the proposed work.

- However, it is not clear what is new when compared to prior approaches that use wearable devices to detect and analyze PD symptoms? A number of such approaches or surveys are listed below

1. Safarpour, Delaram, Marian L. Dale, Vrutangkumar V. Shah, Lauren Talman, Patricia Carlson-Kuhta, Fay B. Horak, and Martina Mancini. "Surrogates for rigidity and PIGD MDS-UPDRS subscores using wearable sensors." Gait & Posture91 (2022): 186-191.

2. Teshuva, Itay, Inbar Hillel, Eran Gazit, Nir Giladi, Anat Mirelman, and Jeffrey M. Hausdorff. "Using wearables to assess bradykinesia and rigidity in patients with Parkinson’s disease: a focused, narrative review of the literature." Journal of Neural Transmission 126 (2019): 699-710.

3. Deb, Ranadeep, Sizhe An, Ganapati Bhat, Holly Shill, and Umit Y. Ogras. "A systematic survey of research trends in technology usage for Parkinson’s disease." Sensors 22, no. 15 (2022): 5491.

- At a minimum, the authors must provide a comparison with some prior approaches that use wearables for this analysis. Both qualitative or quantitative comparisons would be great.

- A description of how other users can use the results of this study would also be useful to understand the utility of the approach.

Reviewer 2 Report

Even when Parkinson’s disease is a relevant topic to get data with sensors, using well-defined hardware and software and comparing those results is not enough to be accepted as a research paper in the Sensors Journal. Therefore, I strongly recommend that authors choose the right journal to discuss their results with reviewers related to this illness. 

Reviewer 3 Report

Thank you for asking me to review this manuscript. The authors have developed a commercially available monitoring system distilling key symptoms and examination findings of parkinsonism into numeric values that can be remotely reported to clinicians. This is an important development and has the potential to revolutionise how we make clinical decisions for people with Parkinson's. Importantly they have compared this to the main clinical/research tool UPDRS in this paper which is a logical approach.

This work is important, however I have recommended a number of points as follows which need addressing prior to a recommendation for publication. One of the main issues is the lack of clarity about the device, specifically how the data points are derived and what they mean which is vital to further review the data presented (both for peer-review and future readers). A few other methodological points need clarifying too. I would find it easier to comment on the actual data if this was better described. 

I would be keen to review the manuscript again with consideration of the following points:

1) The authors have provided a very extensive introduction, however I think it is a bit imbalanced; there is a bit too much on the pathophysiology of Parkinson's and management (which feels superfluous), whilst I would rather have a bit more on background on devices for monitoring in healthcare and as applied to PD.

2) In the patients section of methods, I would like to find out how the patients were recruited. Were they consecutive patients offered the intervention purely out of being in the author's clinic (is this a single clinic, or from across Greece?) or were they selected out of being fit and well and most likely to comply with research?

3) I am not sure what is meant by a recoding session in this case, can this be expanded. Was there one session for each patient? (and presumably one dropped out?)

4) How many expert neurologists did the UPDRS part III assessments, were there several or just one?

5) In the device description, with where monitors are worn, I am not sure what shanks are - potentially the ankle?

6) The method section does not describe how long patients wore the device for and when data was collected - was this at a remote follow up appointment and how long afterwards was this undertaken (on average). Also for people unfamiliar with PDmonitor, it is unclear how the data is collected and summarised for the user - is this an aggregate score over a certain time period? (how long is this epoch?) Is this a mean? The authors describe in a table the basic scores but do not mention the event based montoring referred to later in the results. A whole paragraph explaining each of the PDmonitor parameters and how they are scored would be helpful - eg are they dimensionless arbitrary values? Is tremor based on a mm/cm measurement of average oscillation, or is it time oscillating? Are measures composites of both limbs for tremor or is data individually available for both? I had to visit the PD monitor website to obtain this information - this potentially is extensive and complex but a link to a reference to the more in depth technical detail would be appropriate and at least a more comprehensive description of the measured outcomes by PD monitor in this manuscript. 

7)  line 185 an error has occurred with the referencing software which hasnt been picked up 

8) lines 188-192 appear to be a copy of lines 181-185

9) generally it is a bit confusing which test of agreement is being used where (eg. bland altman vs pearson) - perhaps could try and rewrite so this much clearer what is being used for which value.

10) the comparison of PDmonitor bradykinesia vs clinical rigidity is interesting, but I would like to see if there is a specific link to PDmonitor bradykinesia - is this just because having more of any one PD symptom makes you more like to have another. I recommend commenting on whether other PDmonitor metrics (eg tremor) also correlate to answer this as the authors appear to be inferring that PDmonitor bradykinesia could be used as a surrogate marker of rigidity

11) table 3: the correlation for resting tremor constancy should be presented for the lower limbs as well (similar to severity above in table 3)

12) "Similar to gait, moderate correlation identified on detecting events of 255 posture instability (r=0,46, p<0,05). Despite the moderate correlation in the detection of 256 trunk instability events and gait impairments, very strong correlation was found (r=1) on 257 freezing of gait event detection"

-I do not understand in these sentences what is being compared to what, could it please be rephrased so is really clear what is being assessed and what correlations are being made.  Is this the PDmonitor gait score vs the UPDRS part III postural instability and freezing scores? Or is this the number of freezing episodes detected by the device. However the next paragraph specifically talks about freezing (lines 262-266) so I am a bit confused. Is "freezing of gait event detection" a feature of PD monitor where it records (and counts) freezing episodes? If so this should be the methods section about the device.

13) I do not understand how the device specifically calculates a part 3 score, is this a composite of all parameters measured by PDmonitor? If so this needs to be explained in the methods section

14) Is there a reason bland-altman plots are not performed beyond the tremor and dyskinesia analysis? 

15) Line 296 "The device can vary the level of severity of a symptom or condition over time. In this way, the physician gains a comprehensive picture of symptom fluctuations throughout 297 the day, which underlines the value of continuous monitoring in the management of pa- 298 tients with Parkinson's disease." 

- the first sentence does not really make sense (the device can detect a varying level of severity?) Suggest rephrasing. Additionally the second part really is more discussion rather than belonging in the results section. 

16) Line 299-301 is not really very intuitive: "In Figure 9c the percent of time patients are on slight OFF 299 decreases as the stage of the disease increases (85%, 72% and 25% for H&Y stages 1, 2 and 300 3 respectively)"

- is this trying to state that the severity of OFF time increases with disease stage? 

17) Line 317-218 "In addition to the ability to detect motor symptoms, the PDMonitor® has activity 317 indicators tracking the time a patient is in the state of interest."

-it is not clear what a state of interest is, I believe this may be alluding to activity levels determined by accelerometry, suggest rephrasing.

18) Documentation in the methodology section (or at least a link to this) on how activity level is calculated and presented should be in the methodology section - is this percentage of time moving? How is this differentiated from tremor? (assume use of multiple limb devices to subtract tremor signal). This is an extensive potential are which is only touched upon and could be the subject of an entire seperate paper.

19) line 385 error in referencing software in discussion section

20) References are made to artificial intelligence on a number of occasions. From my understanding whilst PDmonitor may use machine learning algorithims to calculate the parameters presented to clinicians, it is not making high level decisions comparable to human decision equating artificial intelligence. Perhaps if PDmonitor suggested to patients they should take a dispersible madopar or apomorphine injection acutely this could apply - this may be the case but the nature of PDmonitor is not explained enough in the methods section. Could the rationale for artificial intelligence be explained more?

21) Notably several of the authors are affiliated with PD Neurotechnology who have developed the device. This is documented in the conflict of interest statement which is the minimum requirement. I would personally explain in the methods section briefly this link in the name of transparency so this is very clear. Optional but a potential editorial decision.

22) In the methods section I would allude to future work and the next steps. Importantly this should be validated in a larger cohort and scientifically independent of the company. A larger cohort would also cancel out the inter-rater variability seen for the exposure variable which is UPDRS part III. It would be important to assess the impact of clinicians using this device, such as a randomised control trial (use of PDmonitor versus normal care) looking at outcomes such as falls rate, quality of life measures over time and healthcare economic data which would be key to this being adopted by insurance and state healthcare providers going ahead.